# Unbalance Estimation for a Large Flexible Rotor Using Force and Displacement Minimization

**Tuhin Choudhury [1] , Risto Viitala [2] , Emil Kurvinen [1,\*] , Raine Viitala [2] and Jussi Sopanen [1]**

1    Lappeenranta-Lahti University of Technology LUT, School of Energy Systems, Department of Mechanical Engineering, 53850 Lappeenranta, Finland; Tuhin.Choudhury@lut.fi (T.C.); Jussi.Sopanen@lut.fi (J.S.)
2    Aalto University, School of Engineering, Department of Mechanical Engineering, 00076 Espoo, Finland; risto.viitala@aalto.fi (R.V.); raine.viitala@aalto.fi (R.V.)
\*    Correspondence: Emil.Kurvinen@lut.fi; Tel.: +358-505-695-969

**Abstract:** Mass unbalance is one of the most prominent faults that occurs in rotating machines. The identification of unbalance in the case of large flexible rotors is crucial because in industrial applications such as paper machines and roll grinders, high vibrations can adversely affect the quality of the end product. The objective of this research is to determine the unbalance location, magnitude and phase for a large flexible rotor with few measured coordinates. To this end, an established force-based method comprising of modal expansion and equivalent load minimization is applied. Due to the anisotropic behavior of the test rotor, the force method required at least six measured coordinates to predict the unbalance with an error of 4 to 36%. To overcome this limitation, an alternate method, eliminating the use of modal expansion, is proposed. Here, displacements generated by varying the location of a reference unbalance along the rotor axis, are compared to measured displacements to detect the unbalance location. Furthermore, instead of force-based fault models, the minimization of displacements at measured locations determines the unbalance parameters. The test case in this study is the guiding roll of a paper machine and its different unbalance states. The algorithm is tested initially with a simulation-based model and then validated with an experimental set up. The results show that the displacement method can locate the unbalance close to the actual location and it can predict the unbalance magnitude and phase with only two measured coordinates. Lastly, using measured data from 15 measurement points across the tube section of the test rotor, a comparison shows how the selection of the two measured locations affects the estimation accuracy.

**Keywords:** displacement minimization; equivalent load; guiding roll for paper machine; industrial large-scale rotor; modal expansion; unbalance identification

## 1. Introduction

Rotating machinery is an essential component in many modern industries. These machines, operating under real conditions, are susceptible to different faults or defects which are unavoidable. Few such faults are rotor misalignment, rotor bow, transverse crack, and mass unbalance [1]. These faults occur in a rotor because of manufacturing inaccuracies, limited tolerance in parts, inhomogeneous material and imprecise assembly.

Out of all probable rotor faults, mass unbalance is the most frequently occurring source of vibration [2]. Usually, rotating machines go through initial balancing as per their respective class based on standards such as ISO 1940-1 [3]. However, balancing classes allow a certain amount of residual unbalance in the rotor. Such residuals might be insignificant initially. Nevertheless, they may

develop into a larger magnitude due to wear and the accumulation of dirt on rotating parts, creating operational problems which lead to economic or safety related issues.

For heavy machines or rotors operating in unfavorable conditions such as minimum accessibility, high temperatures or pressure, it takes arduous effort and time to replace a damaged rotor. For such critical machines, the continuous monitoring of the system states is necessary to incorporate preventive maintenance and avoid the premature breakdown of the rotor. One of the established methods to do so is to monitor the system's vibration behavior. Over the past few decades, many researchers have contributed to the development of vibration measurement and vibration-based methods for fault diagnosis. More recently, measured vibration signals have been combined with modeling techniques to identify and estimate rotor faults. Some of the modeling techniques use statistical analysis or artificial neural networks to investigate rotor faults [4,5]. However, in the pure sense of model-based identification, a physics-based model enables a direct comparison of dynamic behavior between the simulation model and the actual rotor. The information from this behavioral comparison, correlated with measured vibrations, provides the basis for fault diagnosis [1].

Over the years, many researchers have used modal expansion and the comparison of equivalent forces to determine unbalance among other rotor faults. Platz and Markert [6] have used a model-based method to estimate different fault parameters such as unbalance, transverse crack, rotor-stator rubbing, rotor bow, coupling misalignment, and instability. By performing modal expansion on the measured vibrations, they represented each fault by an equivalent force. These equivalent forces, when compared to the theoretical forces generated by a fault, provided a successful estimation of the fault parameters. Markert et al. [7] have also applied the same method to identify and estimate unbalance in a double-disc rotor-bearing system. In conclusion, they have observed that modal expansion has low accuracy of prediction with few measured coordinates. Sekhar [8] has used a similar approach of using equivalent loads for online identification of cracks in rotors. He modeled a double-disc Jeffcott rotor using the finite element method (FEM), so the transverse crack came across as a local change in the flexibility. The location and depth of the crack were the identifiable parameters in this case. Later, Sekhar [9] studied how the transverse crack and unbalance affected the same rotor simultaneously. He estimated the fault parameters by using two different methods: modal expansion and reduced basis dynamic expansion. He concluded that the latter provided marginally better estimations than modal expansion. Similarly, Jain and Kundra [10] have observed that the occurrence of fault alters the dynamic behavior of the system and they have used modal expansion and equivalent load minimization method for the identification of unbalance and transverse crack. They considered an equivalent load as a fictitious force acting on the undamaged rotor model and successfully validated the numeric simulations with experiments on a double-disc rotor-bearing system. Jalan and Mohanty [11] also used modal expansion combined with a residual generation technique for the online identification of unbalance and misalignment. They used a single-disc rotor-bearing coupling system to predict unbalance in the steady-state condition with an error of 8%. Sinha et al. [12] have estimated both the unbalance and misalignment of a flexibly supported rotating machine from a single run down. They used a theoretical model for rotor and bearing and estimated forces and moments generated by force using measured vibration from the bearing pedestal. For a large test rig, the results showed high sensitivity to rotor modeling errors compared to errors in bearing modeling.

As literature reveals, most of the studies based on modal expansion and equivalent load minimization involved test cases of single-disc or double-disc thin shaft rotors. Therefore, although the method shows good capability in identifying and estimating unbalance, its applicability and accuracy of estimation in large-scale industrial rotors requires further research.

In contrast, some of the previous studies involved the identification of unbalance in complicated industrial rotors using different unique techniques. Bachschmid et al. [13] have identified multiple faults in a finite element (FE)-based rotor model by minimizing the vibration residuals in the frequency domain using least squares fitting. They introduced a residual map to identify the location of the rotor unbalance and the local bow. Furthermore, they validated the numeric experiments by accounting for

modeling error by adding random noise to the bearing coefficients. Pennacchi et al. [14] have used the model-based approach to identify propagating transverse crack in industrial shafts. By using the linear behavior of a horizontally cracked axis, they have identified the crack position by treating it as an external force parameter. Subsequently, Pennacchi et al. [15,16] have advocated that the model-based estimation of unbalance for real machines should include the modal representation of the dynamics of the foundation. For validation, they used experimental data from a 320 MW steam turbo-generator. Furthermore, to establish the robustness of the method in determining fault type, they considered other possible faults and showed that the relative residual of unbalance is minimum, which concurs the prediction of fault type as unbalance. In another two-phase research, Pennacchi [4] has introduced the use of statistical estimators instead of weighted least squares to increase the robustness of estimation in model-based methods. He tested different M-estimators in mechanical problems using simple mass-spring-damper systems. The study shows that Huber's estimator and its modified version takes very few iterations to identify the excitations in the systems. To apply M-estimators in a single unbalance estimation of a real rotor, Pennacchi [17] has considered the 320 MW steam turbo-generator from [16]. He identified the fault location using a color-coded residual map (similar to [13]) of Huber's M estimate. Moreover, he used an iterated re-weighted least squares algorithm to account for measurement noise and modeling error and concluded promising estimations of excitations using Huber's M-estimators. Cedillo and Bonello [18] have taken a non-invasive inverse problem approach to identify unbalance in inaccessible high-pressure rotors. The method uses vibrations at the engine casing and employs least squares to determine the equivalent unbalance distribution in prescribed planes of the rotor. Ocampo et al. [19] have estimated the angular position of unbalance located in an asymmetric rotor by analyzing the polar response plot. They used the amplitude, phase angle and angular velocity of the rotor from four points of the vibration response polar plot and estimated the angular position of the unbalance with a maximum error of 6.7%. In a summarizing review of the existing methods, Lees et al. [1] have studied the different approaches used in model-based identification and concluded that physical models are superior to statistical models due to their ability to imitate the actual physics involved in the system. However, they have also explained the difficulties faced in incorporating non-linear factors like keyways or expansive joints in the model.

More recently, in efforts to improve the accuracy of unbalance prediction using modal expansion, some researchers have modified the technique or combined it with auxiliary methods. For a single-disc rotor, Sudhakar and Sekhar [20] have proposed equivalent load minimization using a modified fault model to improve the unbalance estimation. They compared the estimations by an equivalent load minimization method, its modified version, and a vibration minimization method. For the latter two, the results were promising, even with two measured coordinates. Shrivastava and Mohanty [2] have proposed a method of combining modal reduction using a modified system equivalent reduction expansion process, Kalman filter and recursive least squares method to determine the phase and amplitude of unbalance in a single plane. The process is a suitable alternative to modal expansion once extended to multi-plane unbalance identification. Yao et al. [21] have combined modal expansion along with an inverse problem approach to eliminate the shortcomings related to a low number of measurement points. They studied both single- and double-disc rotor-bearing systems and concluded that the combined method provides more accurate results than purely modal expansion-based estimations.

Overall, the literature survey shows that most researchers have considered the effect of fault as an external force acting on the rotor system. Many of them have used modal expansion to compare model-based unbalance force vector to measurement-based equivalent loads for the full rotor system [1,2,6–11,20,21]. However, these authors have observed that modal expansion has low accuracy of prediction with few measured coordinates. Furthermore, most case studies consisted of a rotor shaft with a single or double disc. In such cases, the unbalance location is clearly identifiable at the disc locations and the equivalent load due to the fault is concentrated locally, and therefore, easier to estimate. In fact, only a few studies have investigated the identification of unbalance in large-scale

industrial rotors [13,16,17] where residual maps and statistical estimators were used to estimate unbalance in combination with 4 measured coordinates (vertical and horizontal at two locations). Furthermore, no studies have been conducted on estimating unbalance in continuous thin-walled rotors, which have industrial applications in paper machines and roll grinders.

The objective of this research is to determine the unbalance location and parameters for a large flexible rotor. To this end, due to the limitations of an established force-based method comprising of modal expansion and equivalent load minimization, an alternate method, eliminating the use of modal expansion, is proposed. In the literature, statistical estimators have been used for identifying the unbalance [16,17] and the least squares method is used at each location to find relative residual and the minimum position in that relative residual surface map gives the location of fault [13]. This study proposes a simplified novel approach where displacements generated by varying the location of a reference unbalance along the rotor axis are compared to measured displacements to detect the unbalance location. Therefore, overall deflection shapes are compared instead of individual amplitudes. When compared to machine learning techniques, which require much pre-processing (such as data filtering, offset removal, data scaling, resampling and data pair preparation) and a large amount of training data, the proposed method requires just the 1X component of the measured signal from only two coordinates as an input which it can directly use for the displacement comparison. Furthermore, instead of force-based fault models, the minimization of displacements determines the unbalance parameters using least squares. As an improvement over existing research, this study uses only two measured coordinates instead of four. As a test case, a guiding roll of a paper machine is considered, which is essentially a large diameter steel tube rotor with thin walls. A sensitivity analysis is conducted for the proposed method to assess its reliability at different speeds, localized modeling error, and measurement noise. Furthermore, the proposed method is tested using measured data from the experimental test rig, and the results are compared to those from the established force-based method. Additionally, using the measured displacements from the 15 measurement points of the test rotor, a comparison shows how the selection of the measured location affects the estimation accuracy.

## 2. Theoretical Background of Model-Based Unbalance Identification Methods

In both methods, the first step is to create a simulation model of the rotor-bearing systems based on FEM. The rotor shaft is discretized into three-dimensional beam elements along its axis of rotation based on Timoshenko's beam theory [22]. Each rotor node has two translational and two rotational degrees of freedom (DOFs). The geometry and physical parameters of the rotor contribute as input to the model. Bearing models include spring and damper elements, where the spring stiffness and damping coefficients are added to the model at nodes of bearing locations. Similar to the bearings, the supports are modeled as spring-damper elements, connecting the bearing to the ground. The simplified rotor-bearing-support model is tested and tuned to match the experimentally derived critical frequencies of the system.

For the numerical simulation cases, the measured vibration is simulated at specific locations for each rotor (usually vertical and horizontal vibrations at two nodes) using a steady-state unbalance response [23]. For the first method, modal expansion of the measured data generates vibrations at all nodes, which are used to calculate the equivalent load for minimization. In the second method, the vibrations at the measured locations are directly subjected to minimization. The following subsections present the theoretical approach and system equations in detail.

### 2.1. Modal Expansion and Equivalent Force Derivation-Based Method

As the literature review reveals, various authors have studied the equivalent load method extensively. Here, the method is only used for a primary validation of the model. Furthermore, it helps to establish some baseline of how the method performs for a large flexible rotor using both numerical simulation and experimental data. However, due to the available literature on the equivalent load method, e.g., [6,7], the details are not included in this current paper.

In short, the method suggests that after the initial grade balancing of a rotor, over a period of operation, the unbalance increases in magnitude, which increases the system vibration. These changes in vibration behavior will lead to the following equation of motion:

$$\mathbf{M}\Delta\ddot{\mathbf{x}}(t) + (\mathbf{C} + \Omega\mathbf{G})\Delta\dot{\mathbf{x}}(t) + \mathbf{K}\Delta\mathbf{x}(t) = \Delta\mathbf{F}(\beta, t) \tag{1}$$

where $\mathbf{M}$, $\mathbf{C}$, $\mathbf{G}$ and $\mathbf{K}$ are the mass, damping, gyroscopic and stiffness matrices of undamaged systems, respectively. These are obtained from the FE model of the rotor. $\Omega$ is the rotor speed. $\Delta\mathbf{F}(\beta, t)$ is the equivalent load due to fault $\beta$. This fault, which is unbalance in this case, results in the change in vibration response represented by $\Delta\mathbf{x}(t)$, $\Delta\dot{\mathbf{x}}(t)$ and $\Delta\ddot{\mathbf{x}}(t)$ in terms of displacement, velocity and acceleration respectively [7].

The vibration responses are measured at certain locations and the full rotor response is obtained using modal expansion [7]. The equivalent loads obtained from Equation (1) are compared to the theoretical formulation of unbalance force to determine the unbalance parameters. Least squares method is used for the optimization [6,9,20,21]. Overall, the method has the limitation that the error in identified fault parameters increases with a decrease in the number of measured coordinates [21,24].

$$\beta(u_r, \alpha_r) \leftarrow \min \sum \left| \Delta\mathbf{F}(\beta, t)_{equivalent} - \Delta\mathbf{F}(t)_{theoretical} \right|^2 \tag{2}$$

## 2.2. Displacement Comparison and Minimization Method

The limitations of modal expansion can be eliminated by comparing displacements at the measured locations instead of equivalent loads to determine the unbalance parameters. In this method, theoretical displacements are obtained from fault models using a steady-state unbalance response. The unbalance force components ($\mathbf{f}_v$ in vertical and $\mathbf{f}_h$ in horizontal) for theoretical fault $\beta$ acting on a node $r$ can be expressed as force vectors:

$$\mathbf{f}_v^r(\beta, t) = \Omega^2 u_r \begin{bmatrix} -\sin\alpha_r & \cos\alpha_r & 0 & 0 \end{bmatrix}^T \tag{3}$$

$$\mathbf{f}_h^r(\beta, t) = \Omega^2 u_r \begin{bmatrix} \cos\alpha_r & \sin\alpha_r & 0 & 0 \end{bmatrix}^T. \tag{4}$$

For a system with $n$ nodes, the force vectors can be assimilated as:

$$\mathbf{F}_v = \begin{bmatrix} \mathbf{f}_v^{1^T} & \cdots & \mathbf{f}_v^{n^T} \end{bmatrix}^T \qquad \mathbf{F}_h = \begin{bmatrix} \mathbf{f}_h^{1^T} & \cdots & \mathbf{f}_h^{n^T} \end{bmatrix}^T. \tag{5}$$

It should be noted that the force vector components $\mathbf{f}_v^n$ and $\mathbf{f}_h^n$ have zero values at all nodes other than the unbalance location. Using these force vectors, the equation of motion for the damaged system can be written as:

$$\mathbf{M}\Delta\ddot{\mathbf{x}}(t) + (\mathbf{C} + \Omega\mathbf{G})\Delta\dot{\mathbf{x}}(t) + \mathbf{K}\Delta\mathbf{x}(t) = \mathbf{F}_v \sin\Omega t + \mathbf{F}_h \cos\Omega t. \tag{6}$$

In Equation (6), the effect of phase angle $\alpha_r$ along with the magnitude of unbalance $u_r$ is incorporated in the force components $\mathbf{F}_h$ and $\mathbf{F}_v$. By using a trial solution of the form:

$$\mathbf{x}(t) = \mathbf{p}_v \sin\Omega t + \mathbf{p}_h \cos\Omega t, \tag{7}$$

the system can be linearized and a solution for displacement can be obtained for a given rotation speed.

$$\begin{bmatrix} \mathbf{K} - \mathbf{M}\Omega^2 & -\Omega(\mathbf{C} + \Omega\mathbf{G}) \\ \Omega(\mathbf{C} + \Omega\mathbf{G}) & \mathbf{K} - \mathbf{M}\Omega^2 \end{bmatrix} \begin{bmatrix} \mathbf{p}_v \\ \mathbf{p}_h \end{bmatrix} = \begin{bmatrix} \mathbf{F}_v \\ \mathbf{F}_h \end{bmatrix} \tag{8}$$

Equations (3)–(8) provide the displacement at the required nodes. Subsequently, these displacements are used to identify the unbalance location and then determine the unbalance parameters as well.

2.2.1. Identifying Unbalance Location

Unlike the equivalent load method, the unbalance location cannot be determined directly based on the measured displacement. Consider an example of a symmetric rotor with an evenly distributed mass, supported by two identical bearings. For the flexible rotor at its first critical speed, the largest displacement usually occurs at the central cross-section of the rotor, whereas the smallest displacements occur at the bearing locations. This is because of the flexibility of the rotor and structural reinforcement at the bearing and support locations. With the consideration of mass unbalance, the displacement amplitude increases at each node. However, the deformed shape of the rotor, for a given speed, almost remains the same. This means that the center of the rotor still has the largest displacement while the bearing locations show the smallest. Therefore, the unbalance location remains unidentifiable.

On the other hand, the displacement at each node from its original static position changes before and after considering the unbalance. The change in displacements at each node, before and after considering the unbalance, can be termed as relative displacement of the node; a parameter that can be studied using the simulation model. This relative displacement is the largest at the location of the unbalance and gradually dissipates with an increase in distance from the point of origin. Irrespective of the deformed mode shape of the rotor, the relative displacement is largest at the unbalance location.

Now consider a test rotor of $n$ nodes with an unbalance whose location and magnitude are unknown. The process flow in Figure 1 displays the different steps involved in identifying the unbalance location.

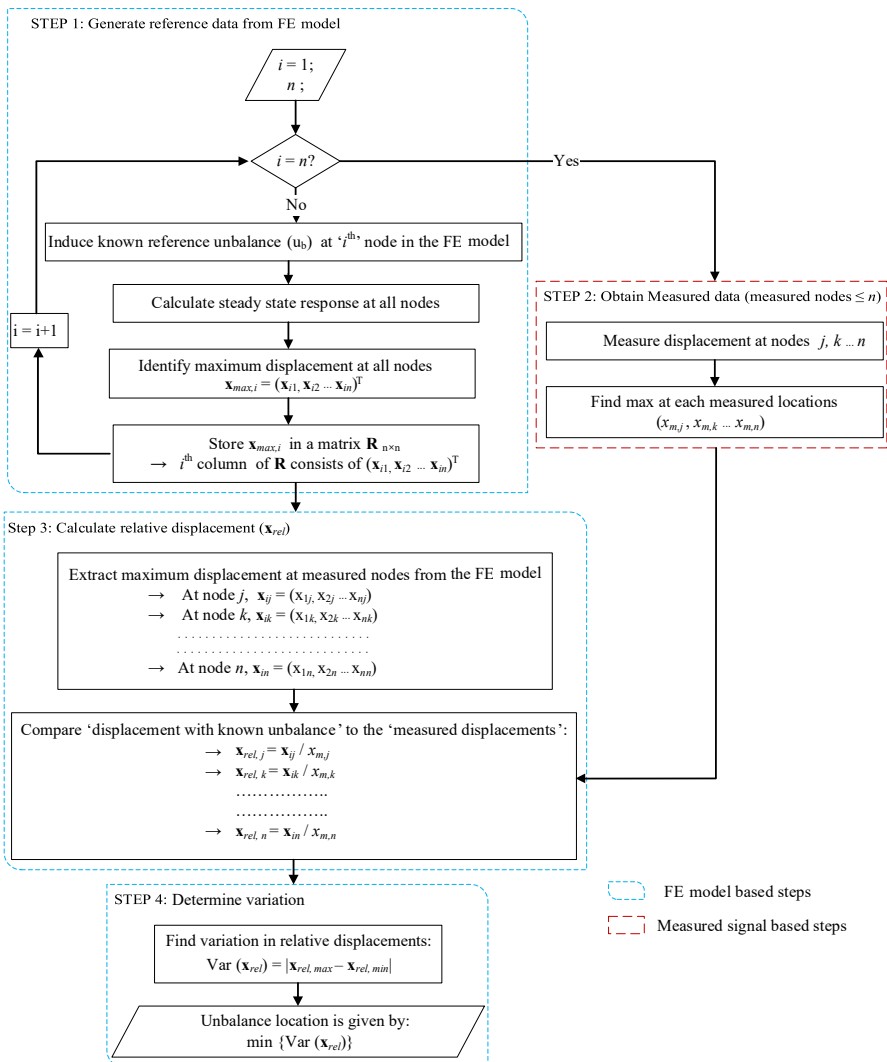

**Figure 1.** Process flow for unbalance detection using two measured coordinates.

The first step is to build a series of reference cases. For this, an unbalance of known magnitude and phase, $u_b$ is set at the first node in the FE model. The steady-state response generates the displacements for the first reference case. For the next reference case, the algorithm generates the response when $u_b$ is at node 2. Similarly, the unbalance location varies consecutively across the length of the rotor, generating a corresponding reference case. From each reference case, only the maximum amplitude of the response at each node is stored. For a reference case with $u_b$ at node $i$, vector $\mathbf{x}_{max,i}$ collects the maximum amplitudes (in either vertical or horizontal direction) at each node as

$$\mathbf{x}_{max,i} = \begin{bmatrix} x_{i1} & x_{i2} & \cdots & x_{in} \end{bmatrix}^T. \tag{9}$$

Each reference case generates $n$ number of displacements using the steady-state response. Therefore, as $u_b$ shifts from nodes 1 to $n$, $n$ number of $\mathbf{x}_{max,i}$ values are generated. These values are stored as columns, eventually forming a matrix $\mathbf{R}$ of size $n \times n$, which provides the required reference data from FE model. Appendix A provides further details on the elements in matrix $\mathbf{R}$. In the meantime, the measured data are obtained for the actual rotor with unknown unbalance at an unknown location. The measured signal is considered to be obtained from at least one measured coordinate from two independent nodes. For example, the displacement in the vertical DOF at each bearing location is measured. However, for a generalized case, the maximum number of measured nodes should be less than or equal to the total rotor nodes. As with the reference case, only the maximum amplitude for each measured DOF is used. For measured locations $j$, $k$ ... $n$, the measured maximum amplitudes are denoted by $x_{m,j}$, $x_{m,k}$ ... $x_{m,n}$ respectively.

The next step compares each reference case to the corresponding measured case. For this, the maximum amplitude at locations $j$, $k$ ... $n$ for each position of $u_b$ are extracted. This means the $j$th, $k$th ... $n$th row of matrix $\mathbf{R}$.

$$\mathbf{x}_{ij} = \begin{bmatrix} x_{1j} & x_{2j} & \cdots & x_{nj} \end{bmatrix}$$
$$\mathbf{x}_{ik} = \begin{bmatrix} x_{1k} & x_{2k} & \cdots & x_{nk} \end{bmatrix}$$
$$\cdots\cdots\cdots\cdots$$
$$\cdots\cdots\cdots\cdots$$
$$\mathbf{x}_{in} = \begin{bmatrix} x_{1n} & x_{2n} & \cdots & x_{nn} \end{bmatrix} \tag{10}$$

Each value of displacement (with known unbalance $u_b$) in $\mathbf{x}_{ij}$, $\mathbf{x}_{ik}$ ... $\mathbf{x}_{in}$ is divided by their corresponding measured displacements, $x_{m,j}$, $x_{m,k}$ ... $x_{m,n}$, to obtain the relative displacements, denoted by $x_{rel}$.

$$\mathbf{x}_{rel,j} = \mathbf{x}_{ij} \;/\; x_{m,j}$$
$$\mathbf{x}_{rel,k} = \mathbf{x}_{ik} \;/\; x_{m,k}$$
$$\cdots\cdots\cdots$$
$$\mathbf{x}_{rel,n} = \mathbf{x}_{in} \;/\; x_{m,n} \tag{11}$$

In Figure 1, as step 3 compares the reference cases with the measured data, both reference data (built in step 1) and measured data (built in step 2) are used as input at this step (more specifically the simulation data is used alone in subsection 1 of step 3 whereas both reference data and measured data are used directly in the second sub-step inside step 3 box). Theoretically, the measured displacements, $x_{m,j}$, $x_{m,k}$ ... $x_{m,n}$ must be similar to the particular column in Equation (10), which has the actual location of unbalance. Therefore, for the relative values in Equation (11), the variation for that particular column must be minimum. As a result, by determining the variation for each column, the column with minimum variation yields the unbalance location.

$$\mathrm{Var}\,(\mathbf{x}_{rel}) = \left| \mathbf{x}_{rel,max} - \mathbf{x}_{rel,min} \right| \tag{12}$$

$$\text{Unbalance location} \leftarrow \min \left\{ \text{Var} \left( \mathbf{x}_{rel} \right) \right\} \tag{13}$$

In this study, the proposed approach is used to identify a single unbalance in a flexible rotor. However, in theory, the method could be applied to any general rotor with multiple unbalances. For a rotor with $n$ discs, $n$ reference unbalance should be considered. A comparison between displacements with all possible combinations of the reference unbalance locations with the measured displacements could provide the actual unbalance locations (or a few possible combinations of locations for multiple unbalance). At the end of the unbalance identification process, the number of relevant locations with minimal variation would be equal to the number of discs.

### 2.2.2. Estimating Unbalance Parameters

Once the unbalance location is known, the next step is to generate displacements fault models. Based on an initial value of the unbalance amplitude and phase, the prediction algorithm goes through Equations (3)–(8) in a loop to generate a displacement fault model. Each iteration provides a displacement output $\mathbf{x}(t)$ obtained from the trial solutions in Equation (7). A comparison between the output displacement and the measured displacement by the least squares method [20] provides the estimated values for the unbalance magnitude and phase. The MATLAB routine 'lsqcurvefit' was used for the optimization.

$$\boldsymbol{\beta}(u_r, \alpha_r) \leftarrow \min \sum \left| \mathbf{x}(t)_{model} - \mathbf{x}(t)_{measured} \right|^2 \tag{14}$$

## 3. Test Rig Description and Modeling

The two different methods for unbalance prediction are tested on a guiding roll of a paper machine which is a large diameter continuous tube with thin walls and hence a more evenly distributed mass system.

### 3.1. Experimental Setup

Figure 2 presents the test setup in which the guiding roll of a paper machine was studied. The test rotor was initially balanced by applying the influence-coefficient method [25] with two balancing planes using the measurement from radial bearing force sensors as feedback. This means that the radial bearing forces are minimized during the balancing process and thus, the orbit of the rotor is nearly circular even if the foundation is anisotropic. Since the rotor was flexible, it could be optimally dynamically balanced only at certain rotating frequency using this balancing method. A rotating frequency 16 Hz was selected as a balancing frequency and same frequency was used also in the measurements. The balancing masses and their phases are presented in Table 1.

**Table 1.** Balancing masses and their phases.

| | Tending End of the Rotor | | | | Driving End of the Rotor | | | |
|---|---|---|---|---|---|---|---|---|
| Balancing masses (g) | 2883 | 1056 | 159 | 0 | 2592 | 499 | 0 | 292 |
| Phases(deg) | 0 | 90 | 180 | 270 | 0 | 90 | 180 | 270 |

The developed unbalance analysis used the lateral displacement of the rotor center point that was measured in this test setup exploiting the four-point method [26]. This method enabled the accurate separation of roundness profile from the center point movement. The measurement was completed with four reflective laser sensors arranged at certain angles according to the method (Figure 2a).

To test the unbalance detection algorithm, three different unbalance cases were considered. The first case included only the residual unbalance that were left after the initial balancing. The rest of the cases included different unbalance masses at the other end of the rotor at different phases.

The unbalance masses and phases for each case are listed in Table 2. For each case, the unbalance masses were attached to the NDE of the rotor just at the end of the tube section (Figure 2b).

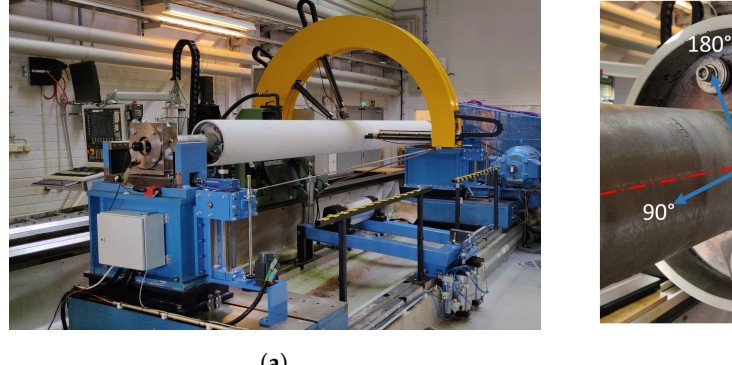 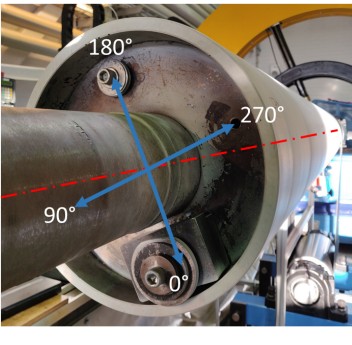

(**a**) (**b**)

**Figure 2.** The final test setup. (**a**) The reflective laser sensors are attached to the yellow arc (**b**) The unbalance masses and the balancing masses attached to the Non-Drive End (NDE) end of the tube section.

**Table 2.** Unbalance mass, eccentricity, magnitude and phase in the measured cases.

| Test Case | Mass (g) | Eccentricity (mm) | Magnitude (kg·m) | Phase (degree) |
|---|---|---|---|---|
| 0 | 0 | 0 | 0 | 0 |
| 1 | 98.6 | 112.5 | 0.011 | 90 |
| 2 | 300.1 | 112.5 | 0.033 | 180 |
| 3 | 497.9 | 112.5 | 0.056 | 270 |

The simulation model was tuned to correspond the actual test setup. For this, the critical speed and subcritical harmonic frequencies were identified by measuring the displacement response of the rotor from the middle section of the rotor. The measurement consisted of acceleration ramp below the critical speed. The results of the response measurement can be seen in Figure 3.

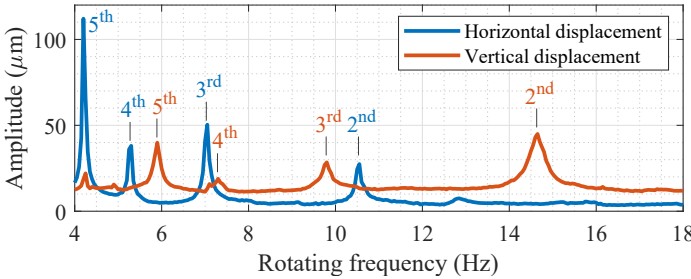

**Figure 3.** Response measurement of the test rotor. The results are separated to horizontal and vertical components.

Figure 3 shows that rotor has anisotropic support since vertical and horizontal directions have their own critical frequencies. The critical frequency for each direction can be calculated from the measured subcritical frequencies; the subcritical frequencies occur at the fractions of the critical frequency such as $1/2, 1/3, 1/4, \ldots$, times the critical frequency, and thus the critical frequency can be solved from each subcritical frequency by multiplying them with their ordinal number. However, the resonance peak at subcritical frequency occurs only if excitation frequency coincides with the critical frequency. Hence, a rotating system must have an excitation that occurs multiple times per revolution to cause resonance. The excitations that occur multiple times per revolution can be e.g., ovality of the bearing inner ring, bending stiffness variation of a rotor or other external source [27]. Table 3 shows the calculated critical frequency and subcritical frequencies of the response measurement.

**Table 3.** Critical speed and subcritical frequencies.

|  | Critical Speeds (Hz) | 2nd Comp.* (Hz) | 3rd Comp. (Hz) | 4th Comp. (Hz) | 5th Comp. (Hz) |
|---|---|---|---|---|---|
| Horizontal | 21.10 | 10.55 | 7.05 | 5.3 | 4.2 |
| Vertical | 29.35 | 14.65 | 9.8 | 7.3 | 5.9 |

\* comp. is short for component.

### 3.2. Modeling Description

Figure 4 describes the main dimensions of the rotor. The guiding roll model is discretized into 24 beam elements along the rotor's length (Table 4). Node 1 is the drive end whereas Node 25 signifies the non-drive end of the rotor. Nodes 2 and 24 are the bearing locations. The support is connected at bearing locations with additional nodes. The support structures contribute to an additional mass of 127 kg each at their respective locations. Furthermore, the overhanging part from the tube end is also considered to be an additional mass point of approximately 6 kg at node 6 and node 20, respectively. Lastly, for the initial balancing of the tube roll, the balancing planes are identified at nodes 6 and 20. The masses used for initial grade balancing in the test rig are added to the balancing planes at either end of the tube section. The additional unbalance masses are attached to the tube end closer to NDE of the roll, which corresponds to node 20.

The foundation stiffness of the simulation model was defined by using foundation design and the corresponding stiffness values obtained from static FE analysis and then matched with the measured peak response frequencies in the horizontal and vertical directions. The corresponding stiffness values in the horizontal and vertical directions are 18 MN/m and 200 MN/m, respectively. Damping of the support is fine-tuned according to the shape of the response curves and the damping ratios are evaluated as 2% in the vertical direction and 1.3% in the horizontal direction. The corresponding first and second critical speeds obtained in the model are 20.91 Hz and 29.31 Hz which are quite close to the measured values with relative error of 0.9% and 0.1% respectively. Figure 5 shows the first horizontal and vertical modes obtained from the model of the rotor.

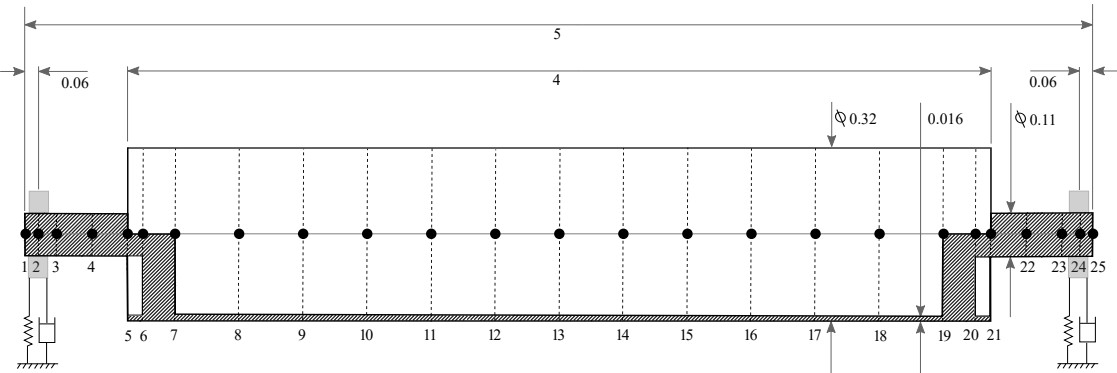

**Figure 4.** A sketch of the Paper machine's steel tube roll with FE discretization. All dimensions are in meters.

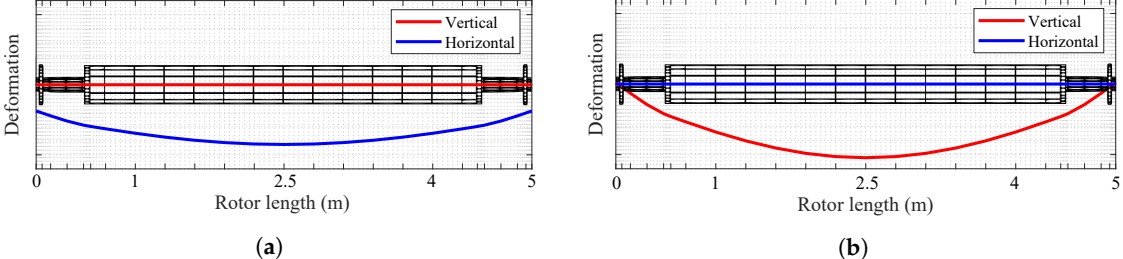

**Figure 5.** 1st critical speeds from the simulated model (**a**) Horizontal direction (20.91 Hz) (**b**) Vertical direction (29.31 Hz).

**Table 4.** Parameters of tube roll rotor-bearing system.

| | |
|---|---|
| *Rotor properties* | |
| Density | 7764 kg/m$^3$ * |
| Poisson's ratio | 0.3 |
| Young's modulus | $1.99 \times 10^{11}$ Pa * |
| Rotor mass (measured) | 719.72 kg |
| *Bearing properties* | |
| *Bearing stiffness coefficient* | |
| Vertical | $2.5 \times 10^8$ N/m |
| Horizontal | $4.3 \times 10^7$ N/m |
| *Bearing damping coefficient* | |
| Vertical | $2.5 \times 10^5$ Ns/m |
| Horizontal | $4.3 \times 10^4$ Ns/m |

* The density and Young's modulus are calculated by tuning the model to match the actual free-free frequencies of the tube roll.

## 4. Numerical Simulation

This section demonstrates the accuracy of each method in identification and estimation of unbalance for both of the case studies for a series of simulated test runs with different unbalance locations and parameters.

### 4.1. Unbalance Estimation Using Equivalent Load Minimization and Modal Expansion

First, the unbalance location, magnitude, and phase are estimated for individual test rotors using equivalent load minimization and modal expansion.

### 4.1.1. Identifying Unbalance Location

Theoretically, the unbalance forces act only on the node where the unbalance is located and therefore, the index of the active forces directly provide the unbalance location. However, the equivalent load in the current simulations is derived based on modal expansion. Therefore, the node with the highest magnitude of equivalent load is considered to be the location of unbalance.

However, it seems that depending on the structure of the rotor and the location of the unbalance, the maximum equivalent load does not always provide the correct unbalance location. This is particularly true for the current test rotor which has a highly asymmetric bearing and support properties, especially with lower number of measured coordinates.

Figure 6 shows the equivalent load distribution for the tube roll for one cycle of steady-state response at 16 Hz rotation speed when an identical unbalance is induced at node 20 for two different sets of measured nodes. With four measured coordinates (Nodes 6 and 19), the equivalent load method incorrectly predicts the unbalance location at the center of the roll (node 13). The method requires a minimum of six measured coordinates (nodes 6, 14, 19) to predict the unbalance location correctly at node 20.

However, in the case of a continuous rotor, such as the tube roll where the mass is more evenly distributed, the unbalance can occur anywhere along the length of the tube. Therefore, identifying the unbalance location is more challenging, and as Figure 6 shows, the modal expansion and equivalent load method was unable to predict the unbalance location correctly. Therefore, this method might not be suitable for unbalance detection in continuous rotors.

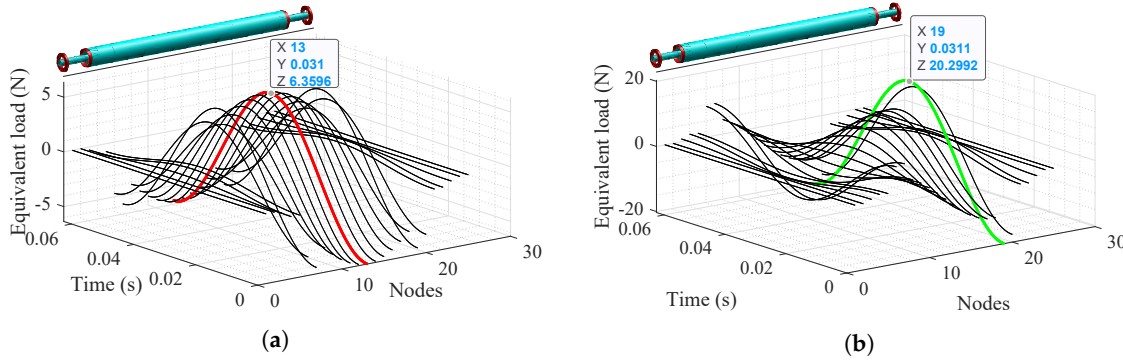

**Figure 6.** Detecting the unbalance location by identifying the position of the maximum equivalent load in the vertical direction for the tube roll with different sets of measured coordinates: (**a**) With 4 DOFS (**b**) With 6 DOFS.

#### 4.1.2. Estimating Unbalance Parameters

The unbalance parameters consist of its amplitude and phase. Three different scenarios with varying unbalance magnitude and phase are considered for test cases. For the force method, two different sets of measured coordinates are used (Table 5). The vibration signals at these coordinates represent the measured data in the simulated cases. The results reflect the stability of the algorithm in estimating a range of unbalance magnitude and phase values.

**Table 5.** Selection of nodes for corresponding numbers of measured coordinates in the tube roll.

| Number of Measured DOFs | Measured Nodes (Transverse Vibrations at Each Node = 2 DOFs) |
|---|---|
| 6 | 6, 14, 19 |
| 4 | 6, 19 |

Once the unbalance location is identified correctly, the shifting and summation of all the equivalent load to the unbalance location is required [28]. This is because the paper machine's tube roll is a continuous rotor with a relatively more even distribution of mass along its axis. Therefore, the equivalent load is spread across a few nodes instead of being isolated at the unbalance location and must be included in the calculation.

**Table 6.** Unbalance magnitude and phase estimation for the steel tube roll using equivalent load minimization using six and four measured coordinates.

| Test No. | No. of Measured DOFs | Unbalance Location (Node) | | Unbalance Magnitude (kg·m) | | | Unbalance Phase (degree) | | |
|---|---|---|---|---|---|---|---|---|---|
| | | Actual | Identified | Actual | Estimated | Error % | Actual | Estimated | Abs. Error |
| 1 | 6 | 20 | 19 | 0.011 | 0.006 | 41.34 | 90 | 90.5 | 0.5 |
| 2 | 6 | 20 | 19 | 0.033 | 0.020 | 41.37 | 180 | 178.6 | 1.4 |
| 3 | 6 | 20 | 19 | 0.056 | 0.032 | 41.36 | 270 | 270.5 | 0.5 |
| 1 | 4 | 20 | 13 | 0.011 | 0.005 | 58.88 | 90 | 90.5 | 0.5 |
| 2 | 4 | 20 | 13 | 0.033 | 0.014 | 58.91 | 180 | 178.6 | 1.4 |
| 3 | 4 | 20 | 13 | 0.056 | 0.023 | 58.89 | 270 | 270.5 | 0.5 |

Even with the shifting and summation of the equivalent load, Table 6 shows that the accuracy of the estimation of the unbalance magnitude is about 32.68% for six, and to 54.12% for four measured DOFs. This high inaccuracy is probably because of the complexity and continuous design of the tube roll rotor. The method includes the approximation of vibrations at unknown locations by performing a modal expansion over the measured signals. These approximated vibrations are used to calculate the equivalent load, which in theory, is identical to the theoretical unbalance forces. However, due to the more distributed mass of the tube, some of the theoretical force remains unaccounted for in the approximation of modal expansion. This leads to a relatively inaccurate estimation of the unbalance

magnitude and phase. Another key factor for the high inaccuracy could be the highly asymmetric parameters of the bearing and supports.

*4.2. Unbalance Estimation Using Displacement Comparison and Minimization*

Next, the displacement comparison method is tested using only two measured coordinates. These measured DOFs are located at the node 6 and node 19. The measured signal is simulated for the vertical displacements. The reason for choosing the vertical direction is that the support is considerably softer in the horizontal direction than in vertical (18 MN/m horizontal and 200 MN/m vertical support stiffness); only the shaft is defining the dynamics in the vertical direction, whereas in the horizontal direction, the looseness in the support influences also to the dynamics. Therefore, the vertical response is more linear for this particular test case and provides a more realistic 1X response due to unbalance only. Since with the proposed algorithm, only two coordinates are required (assuming that measurement sensors are operational or mounted in any one direction only), the vertical displacements are used in this study.

4.2.1. Identifying Unbalance Location

In this method, the unbalance location is determined in four steps where measured displacements from two DOFs combined with model-based reference data identifies the required location. Figure 7 describes an example case to elaborate on the process flow. Here, the tube roll is induced with an unbalance (magnitude 0.011 kg·m and phase 180 degree) at node 20. The unbalance parameters and location are considered to be unknown for testing the method.

The first step is creating a set of reference cases using a known unbalance ($u_b$). Here, $u_b$ is considered to have a magnitude of 1 kg·m and 0 degree phase (the parameter values can be chosen arbitrarily). The position of $u_b$ is shifted consecutively from node 1 to node 25 in the FE model of the tube roll. For each location of $u_b$, the steady-state unbalance response generates the displacement in all the nodes. Out of these values, only the maximum amplitudes ($x_{ref}$) at the locations corresponding to measured DOFs (node 6 and node 20) are extracted and plotted in Figure 7a). The values are stored in a 2 × 25 matrix as extracted reference data from the model. The mathematical representation of this step is provided shown in Appendix A where $\mathbf{R}_{extract}$ represents the values plotted in Figure 7a).

In step 2, the measured displacement signals ($x_{meas}$) from node 6 and node 20 are simulated for the tube roll for a single period of rotation (Figure 7b). It is entirely possible to simulate the measured data for multiple rotations. However, that would only contribute to a lot of unnecessary computation time without any changes in the measured data since the rotor speed is constant. The maximum amplitude values are identified for each measured signal ($x_{m,j}$ and $x_{m,k}$).

The next step compares the maximum amplitude with the known unbalance $u_b$ ($x_{ref}$) obtained from the FE model to the measured maximum amplitudes ($x_{m,j}$ and $x_{m,k}$). Figure 7c shows the resulting plots of relative displacements ($x_{rel}$) when each individual $x_{ref}$ for nodes 6 and 20 are divided by their corresponding measured maximum values ($x_{m,j}$ and $x_{m,k}$, respectively).

Since this example considers only two measured signals, there is no requirement to calculate the maximum and minimum values of $x_{rel}$ at each location to determine the variation. It can be simply obtained by subtracting the $x_{rel}$ of one measured data from another at each location. Figure 7d shows the final step where the variation is determined by subtracting the $x_{rel}$ at nodes 6 and 19 obtained from Figure 7c. The node with minimum variation gives the unbalance position. In this case, the variation appears to be close to zero at node 1, 20 and 25 but upon closer inspection, the minimum variation occurs at node 20 which corresponds to the correct location of the induced unbalance. Although the method looks susceptible to a wrong prediction due to the very low variation values at node 1 and 25, it should be noted that for such a large rotor with insignificant overhung, the variation in the overhung region is very small regardless of where the unbalance is located in the tube section. Regardless of that special case, the method has correctly predicted the unbalance at node 20 even when tested at over 1st and 2nd critical speeds (Figure 8) which shows the robustness of the location identification for an

accurate model. Furthermore, it is also possible to eliminate those overhung locations because based on the tube shape design of the rotor with very small overhung regions, the unbalance is predominantly distributed along the length of the tube section of the rotor [29].

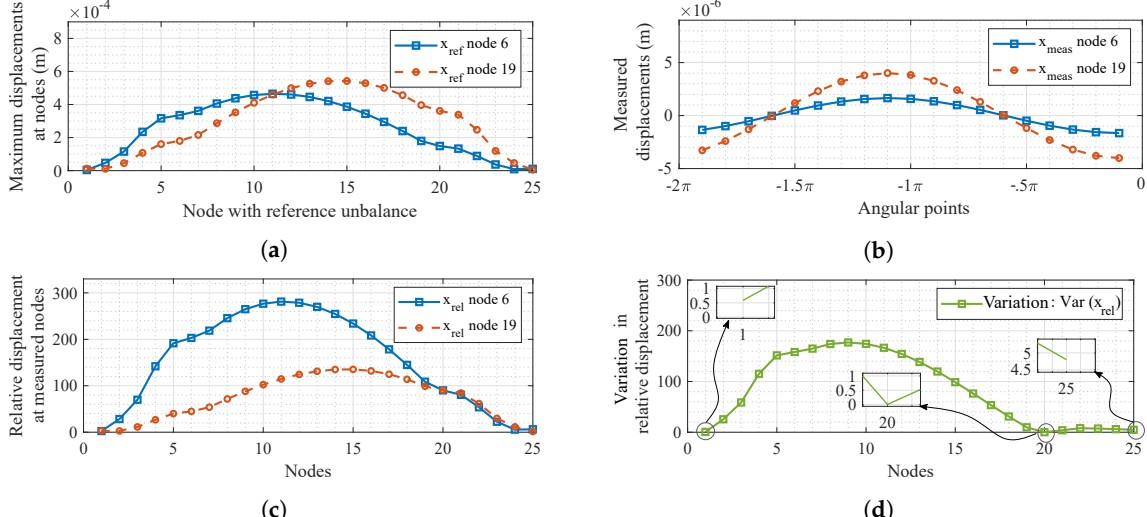

**Figure 7.** Identification of the unbalance location using the displacement comparison method for the tube roll. The induced unbalance (magnitude of 0.056 kg·m and 0 degree phase) is located at node 20. Subplots (**a**) The maximum amplitude ($x_{ref}$) of each node for individual reference cases where the unbalance location varies from node 1 to node 25 consecutively in the FE model. (**b**) Measured displacements ($x_{meas}$) at bearing locations with the maximum amplitudes identified. (**c**) Relative displacement ($x_{rel}$): reference cases ($x_{ref}$) divided by max. measured displacements. (**d**) Variation at each node: the minimum value gives the unbalance location.

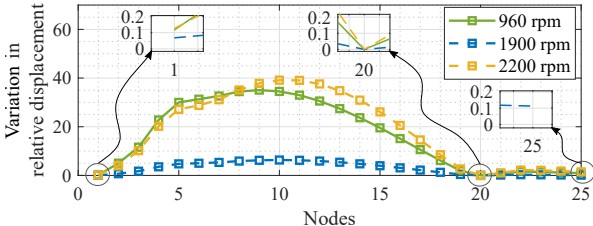

**Figure 8.** Location prediction accuracy test at three different speeds for induced unbalance at node 19.

Once the process flow is clearly established, the location prediction accuracy of the proposed method is tested for the tube roll at three different speeds. The first speed is set at 960 rpm which is just below the first critical speed (horizontal cylindrical mode). The second and third speeds for the simulation are 1900 and 2200 rpm which are just above the first vertical and second horizontal critical speeds of the rotor. Figure 8 shows that the algorithm can predict the location correctly for each of the three speeds. For the 960 and 2200 rpm which are closer to horizontal resonances, the variation is higher and for the 1900 rpm which is closer to vertical resonance, the variation values are smaller. However, regardless of the deformation shapes and their corresponding variation, the overall pattern of identifying the unbalance location remain quite similar.

### 4.2.2. Identifying Unbalance Parameters and Sensitivity Analysis

Once the algorithm can identify the unbalance location, it is tested to estimate unbalance parameters. The induced unbalance values are similar to the ones used in testing the equivalent load minimization method. However, all tests include simulations of measured signals from two DOFs only.

The unbalance parameters are detected using two simulated vibration signals, representing the measured vertical vibrations at node 6 and node 19. Table 7 shows the estimation results for different values of unbalance for the tube roll rotating at 960 rpm. The prediction accuracy is quite good and consistent across a considerable range of values of the unbalance magnitude and phase.

**Table 7.** Estimation results for different values of unbalance magnitudes and phases induced in the paper machine's tube roll using the displacement minimization method.

| Test Cases | No. of Measured DOFs | Unbalance Location (Node) | | Unbalance Magnitude (kg·m) | | | Unbalance Phase (degree) | | |
|---|---|---|---|---|---|---|---|---|---|
| | | Actual | Identified | Actual | Estimated | Error % | Actual | Estimated | Abs. Error |
| 1 | 2 | 20 | 20 | 0.011 | 0.011 | 0.00 | 90 | 90 | 0 |
| 2 | 2 | 20 | 20 | 0.033 | 0.033 | 0.00 | 180 | 180 | 0 |
| 3 | 2 | 20 | 20 | 0.056 | 0.056 | 0.00 | 270 | 270 | 0 |

However, unlike the simulation-based measured data, real measurements always include a certain amount of noise. Furthermore, the rotor model can have some missing or incorrect information, which would lead to modeling errors. Therefore, to test how the method reacts to different sources of error, a sensitivity analysis is performed (Table 8). The modeling errors include 1%, 2% and 5% of localized error in mass. These errors are induced at one element of the tube section of the rotor model. The measurement error of 5%, 15% and 25% are added by incorporating multiplicative noise of the form 'nX' where 'n' is uniformly distributed random noise and 'X' is the clean simulated signal. Similar to the unbalance location identification test, each case is tested at two different speeds: 960 rpm which is the operational speed and 2200 rpm (36.6 Hz) which is just above the second horizontal resonance speed (conical mode) at 35 Hz. Overall, the method seems to work well with modeling errors and measurement noise irrespective of the rotor speeds. Although it should be noted that the method is highly sensitive to stiffness coefficients. This is probably because of the very high stiffness of the rotor and the foundation where even 1–2% change in the local stiffness coefficient leads to exponential changes in the stiffness matrix.

**Table 8.** Estimation of unbalance parameters in the paper machine's tube roll with different types of modeling and measurement-based errors.

| Speeds (rpm) | Test Cases | Unbalance Parameters | Actual Values | Measurement Noise | | | Modeling Error | | |
|---|---|---|---|---|---|---|---|---|---|
| | | | | 5% | 15% | 25% | 1% | 2% | 5% |
| 960 | 1 | magnitude | 0.011 | 0.011 | 0.012 | 0.013 | 0.011 | 0.011 | 0.012 |
| | | phase | 90 | 90.03 | 90.74 | 90.64 | 90 | 90.82 | 90.53 |
| | 2 | magnitude | 0.033 | 0.035 | 0.038 | 0.042 | 0.033 | 0.033 | 0.034 |
| | | phase | 180 | 179.99 | 179.35 | 179.64 | 97.8 | 98.4 | 101.6 |
| | 3 | magnitude | 0.056 | 0.058 | 0.064 | 0.070 | 0.056 | 0.056 | 0.057 |
| | | phase | 270 | 269.79 | 271.26 | 273.10 | 270 | 269.7 | 269.4 |
| 2200 | 1 | magnitude | 0.011 | 0.011 | 0.012 | 0.013 | 0.010 | 0.010 | 0.011 |
| | | phase | 90 | 90.07 | 90.38 | 90.54 | 90 | 90.21 | 90.64 |
| | 2 | magnitude | 0.033 | 0.035 | 0.038 | 0.042 | 0.033 | 0.032 | 0.035 |
| | | phase | 180 | 180 | 180.08 | 180.19 | 180 | 180.53 | 180.64 |
| | 3 | magnitude | 0.056 | 0.058 | 0.064 | 0.070 | 0.055 | 0.054 | 0.058 |
| | | phase | 270 | 270.04 | 269.82 | 269.47 | 270 | 270.07 | 270.14 |

## 5. Experimental Verification

### 5.1. Obtaining Measured Signal and Pre-Processing

The unbalance algorithm was verified by experimental measurement data. The acquired displacement data was measured from 15 different cross-sections along the rotor in four different measurement cases. The measured points were selected to correspond the nodes in the simulation model. Measurement procedure for each case proceeded as follows:

1. The operation speed of the roll was set at 960 rpm (16 Hz). The acceptable speed range for this rotor is 4–18 Hz.
2. Measuring probes were driven to the first measuring point.
3. 100 revolutions of the rotor center point movement were measured from the first measuring point.
4. Measuring probes were driven to the next measuring point.
5. Steps 3 and 4 were repeated until the measurement is conducted also in the last measuring point.

100 revolutions of data were acquired from 15 different measuring points. To eliminate noise from the signal, time synchronous averaging (TSA) [30] were applied in post processing. TSA eliminates noise from the harmonic signal  and filters out phenomena that do not occur  every revolution. After the TSA, it is beneficial to present measured frequencies in relation to the rotating frequency of the rotor that facilitates excitation identification. To apply the TSA method, phase locked signal is needed to provide an external trigger for signal acquisition. In this study, the external trigger was an encoder attached to the other end of the rotor that ensures signal acquisition exactly from the same phase. The resolution of the encoder was 1024 pulses per revolution and thus, 1024 displacement points were measured during one revolution. The coordinates of the measured system were correlated with the simulated case to ensure correct phase and directional consideration. After the noise filtering, the 1X component of the signal was extracted using Fast Fourier Transform (FFT). Figure 9a,c show the overall vertical displacement signal measured at node 19, the FFT amplitudes and the extracted 1X component for the initial case without any additional mass. Figure 9b,d shows the details but with approximately 0.5 kg unbalance mass at about 0.1125 m radial distance from the rotor centerline. These signals for each unbalance case was subtracted from the massless case to obtain the residuals which were used in the predictive least squares algorithm.

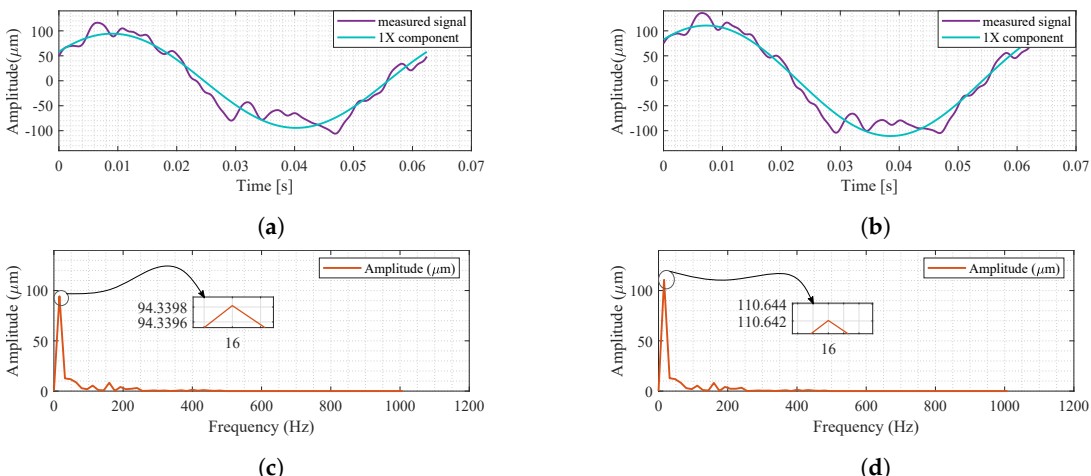

**Figure 9.** Extraction of 1X component from **a** measured signal using FFT. (**a**,**c**) are for the case without any additional mass and (**b**,**d**) show the process for the case with 0.056 kg·m unbalance.

### 5.2. Identification and Estimation of Unbalance

Similar to the simulation studies in the previous sections, the three different test cases are subjected to the force method and then the displacement method for unbalance identification.

### 5.2.1. Force Method

For the force method, two sets of measured coordinates are used: 6 DOFS (nodes 6, 14 and 19) and 4 DOFS (nodes 6 and 19). Table 9 shows how the predictions worked for the three different cases of induced unbalance. The results show that there is almost twice as much error with 4 measured DOFs when compared to 6 DOFs. Moreover, there is a consisting increase in error as the unbalance magnitude increases for a given number of measured DOFs. This is probably due to nonlinearity in the unbalance response for the actual rotor which is not accurately captured in the simulations.

**Table 9.** Unbalance magnitude and phase estimation for the steel tube roll using equivalent load minimization using six and four measured coordinates.

| Test Cases | No. of Measured DOFs | Unbalance Location (Node) | | Unbalance Magnitude (kg·m) | | | Unbalance Phase (degree) | | |
|---|---|---|---|---|---|---|---|---|---|
| | | Actual | Identified | Actual | Estimated | Error % | Actual | Estimated | Abs. Error |
| 1 | 6 | 20 | 19 | 0.011 | 0.012 | 7.44 | 90 | 95.1 | 5.1 |
| 2 | 6 | 20 | 19 | 0.033 | 0.029 | 12.71 | 180 | 167.4 | 12.6 |
| 3 | 6 | 20 | 19 | 0.056 | 0.034 | 38.61 | 270 | 274.9 | 4.9 |
| 1 | 4 | 20 | 13 | 0.011 | 0.007 | 34.52 | 90 | 89.1 | 0.9 |
| 2 | 4 | 20 | 13 | 0.033 | 0.021 | 36.08 | 180 | 171.3 | 8.7 |
| 3 | 4 | 20 | 13 | 0.056 | 0.024 | 55.81 | 270 | 271.2 | 1.2 |

### 5.2.2. Displacement Method

For the displacement method, only two measured coordinates are used and Table 10 shows the prediction results for the three different test cases.

**Table 10.** Estimation results for different values of unbalance magnitudes and phases induced in the paper machine's tube roll. The displacement minimization method is used for the estimation, and each case has two measured coordinates.

| Test Cases | No. of Measured DOFs | Unbalance Location (Node) | | Unbalance Magnitude (kg·m) | | | Unbalance Phase (degree) | | |
|---|---|---|---|---|---|---|---|---|---|
| | | Actual | Identified | Actual | Estimated | Error % | Actual | Estimated | Abs. Error |
| 1 | 2 | 20 | 14 | 0.011 | 0.010 | 3.95 | 90 | 91.3 | 1.3 |
| 2 | 2 | 20 | 19 | 0.033 | 0.040 | 20.21 | 180 | 171.8 | 8.1 |
| 3 | 2 | 20 | 19 | 0.056 | 0.054 | 2.66 | 270 | 269.8 | 0.1 |

Compared to the force method, the displacement method estimated the unbalance magnitude with quite high accuracy. For cases 2 and 3, the method has identified the unbalance location close to the actual plane of unbalance. For case 1, the location prediction is incorrect. This could be due to the relatively very small amount of unbalance mass for case 1 (approx. 0.013% of the rotor mass). Therefore, even with the added unbalance in case 1, the relative change in deflection is higher at the middle of the rotor than at the actual location of the unbalance, thus leading to the incorrect prediction of unbalance location at node 13. Furthermore, the accuracy of the method is the lowest for case 2. Figure 10 provides more insight on the reason behind the different accuracy for the three cases. The figure shows that for case 1, the measured deflection shapes are relatively more irregular than the other two cases. Furthermore, the simulated deflection shapes are not able to replicate the pattern with precision. Due to the inability of the model to follow the fluctuations in the measurement for case 1, the predicted location of the unbalance is incorrect. Coincidentally, with the location being predicted at node 14, the magnitude and phase are predicted with quite high accuracy (96%). However, with the assumption that the location was predicted correctly at node 19, the estimation accuracy for case 1 drops to approximately 40%. Therefore, for this particular case, the accuracy in magnitude is a trade off with the prediction of unbalance location. For case 2, the deflection shape from the simulation model is closer to the measured deflection along the rotor length. Therefore, the algorithm can predict the unbalance location and parameters with better accuracy (67%). For case 3, the deflection shapes have a better match and therefore, the prediction accuracy is quite high (91%) along with the correct prediction of unbalance location.

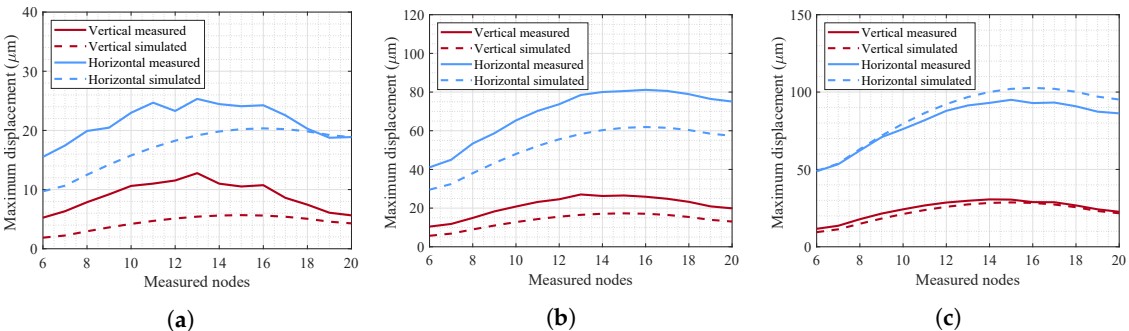

(a)       (b)       (c)

**Figure 10.** Comparison of deflection shape in the tube section of the guiding roll for (**a**) Case 1 (**b**) Case 2 (**c**) Case 3, using maximum displacement amplitudes from measurement and simulated model

5.2.3. Estimation for Different Combinations of Two Measured Nodes Using Displacement Method

Along with the number of measured coordinates, the selection of the location of those measured DOFs plays a key role as well in the prediction accuracy for both methods. Typically, it might seem that selecting all measured DOFs closer to the unbalance location would yield a very accurate estimation. However, practically that is not always the case. In this particular study, measurements from 15 different points were available. Therefore, Figure 11 shows how the predictions are for different combinations of two measured nodes.

For case 1, almost all the combinations estimate the unbalance quite well. Still, measurements from symmetrically opposite points on either side of the rotor yield the best results. The outputs from case 2 are quite inconsistent in comparison with the other cases. The estimations are quite good when one measurement point is around nodes 6 to 10 and the other location is node 14 to 18. For case 3, similar to case 1, high accuracy predictions are obtained when measurement points are closer to each end of the tube section. Overall, the stem plots suggest that the best predictions are obtained when the selected measured nodes are symmetrically opposite and cover higher vibration locations. Furthermore, the comparison shows that in practical applications, the displacements near the tube ends would lead to better prediction accuracy.

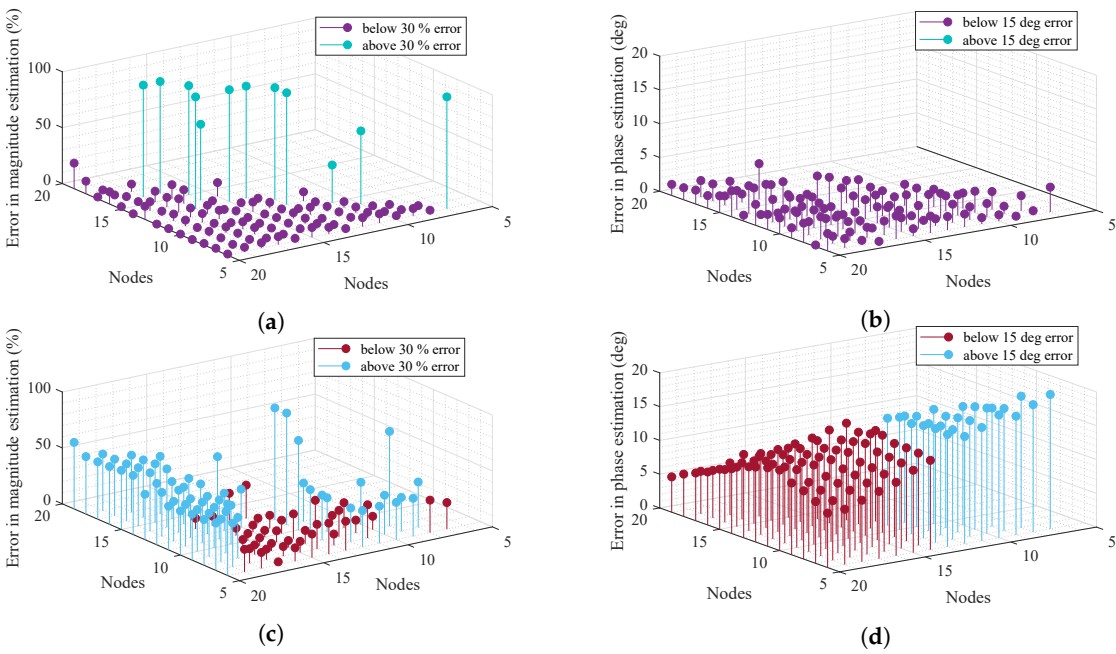

(a)       (b)

(c)       (d)

**Figure 11.** *Cont.*

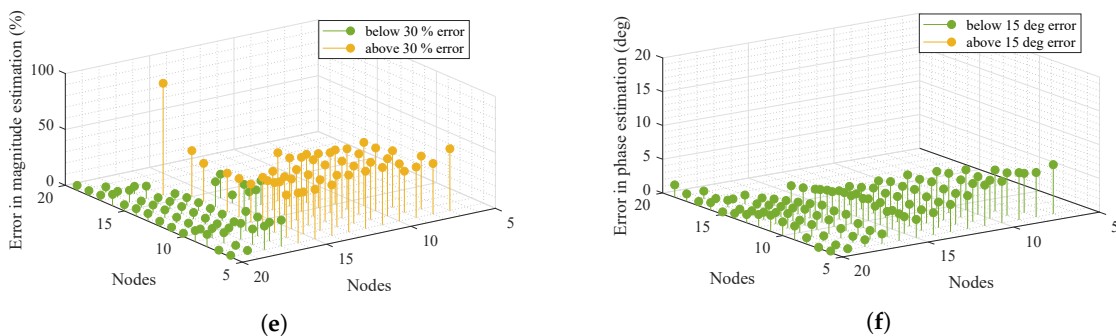

**Figure 11.** Error is estimation of unbalance (**a**) magnitude and (**b**) phase for case 1, (**c**,**d**) for case 2, and (**e**,**f**) for case 3 with different combinations of two measured nodes from the experimental setup.

## 6. Conclusions

The objective of this paper was to develop a method that can identify unbalance location and estimate its parameters, for a large flexible rotor commonly known as guiding roll for paper machine. To establish a base line prediction for reference, first a force-based method including modal expansion, equivalent load minimization and least squares optimization is tested. For this method, predictions were good with 6 measured DOFs but the accuracy dropped with 4 measured DOFs. The alternative proposed method compared displacements instead of equivalent loads to identify the locations and estimate the unbalance parameters in the tube roll. Both numerical simulations and experimental results with varied range of unbalance values showed better predictions for the displacement-based method with fewer measured DOFs when compared to the established force method. Furthermore, the displacement method showed good accuracy and precision when tested in simulations with variable speeds, measurement noise and modeling error. Lastly, using the available measurements from 15 points, a comparison showed how the selection of measurement location affects the predictions for each of the cases. In most cases, selecting measured nodes at symmetrically opposite locations yield the best results. Overall, the proposed displacement comparison method shows reasonably good robustness in unbalance prediction even for real world applications, considering it used only two measured coordinates to locate and estimate unbalance for a 700 kg industrial-scale rotor. Once the unbalance magnitude and phase are known using the proposed algorithm, operators can choose the balancing planes to proceed with the operational speed balancing of the flexible rotor. For the test rotor, the operational speed of 16 Hz is slightly below the first critical speed (21.1 Hz). Therefore, the operators can select either a single [31] or three balancing plane [32] and instead of using trial masses, the balancing masses can be directly obtained using the previously estimated existing unbalance magnitude and phase. Evaluating the hypothesis about how well the balancing in one or three planes would work can be considered to be an objective for future work. Another potential topic for future research could be to expand the proposed method to cases with multiple unbalances. The proposed method has the advantage that it can work for rotating machines where only two sensors are available or functional in any single direction. One limitation of the method is that for systems with multiple faults, the estimations might be incorrect unless the existing faults are increasing the 1X response only, which then could be accounted for as additional unbalance and compensated with additional masses. At this current stage, the algorithm is more applicable in the design phase and early operation phases, when the system can be identified well.

**Author Contributions:** Funding acquisition, J.S. and R.V. (Raine Viitala); Investigation, T.C., R.V. (Risto Viitala), E.K.; Methodology, T.C., E.K., R.V. (Risto Viitala) and J.S.; Project administration, T.C., R.V. (Risto Viitala) and E.K.; Software, T.C., E.K. and J.S.; Supervision, J.S. and R.V. (Raine Viitala); Writing—original draft, T.C., R.V. (Risto Viitala), E.K., R.V. (Raine Viitala), J.S.; Writing—review & editing, T.C., R.V. (Risto Viitala), E.K., R.V. (Raine Viitala), J.S. All authors have read and agreed to the published version of the manuscript.

**Funding:** This work was supported by Academy of Finland [grant number 313676 and 313675].

**Conflicts of Interest:** The authors declare no conflict of interest.

## Appendix A

Appendix A elaborates how the reference matrix **R** is created and how values are extracted from **R**. To start with, the known reference unbalance, $u_b$ is placed at node 1 of the rotor model with $n$ nodes. The maximum amplitude for this case is obtained at each node. These maximum amplitudes are stored in $\mathbf{x}_{max,\,1}$ by using Equation (9). Next, the unbalance $u_b$ is shifted to node 2, and maximum amplitudes for each node of the model are stored in $\mathbf{x}_{max,\,2}$. The process is repeated for all $n$ nodes in the rotor.

$$\text{Reference } u_b \text{ is at node 1,} \quad \mathbf{x}_{max,1} = \begin{bmatrix} x_{11} & x_{12} & \cdot & x_{1j} & \cdot & x_{1k} & \cdot & x_{1n} \end{bmatrix}^T$$

$$\text{Reference } u_b \text{ is at node 2,} \quad \mathbf{x}_{max,2} = \begin{bmatrix} x_{21} & x_{22} & \cdot & x_{2j} & \cdot & x_{2k} & \cdot & x_{2n} \end{bmatrix}^T$$

$$\cdots \cdots \cdots \cdots \cdots$$

$$\text{Reference } u_b \text{ is at node n,} \quad \mathbf{x}_{max,n} = \begin{bmatrix} x_{n1} & x_{2n2} & \cdot & x_{nj} & \cdot & x_{nk} & \cdot & x_{nn} \end{bmatrix}^T$$

The values of $\mathbf{x}_{max,\,1}$ are marked by the thick-lined box just to demonstrate how the values are stored in the reference matrix. The maximum amplitudes from each reference cases are stored as individual columns to build the matrix of reference data **R** (size $n \times n$), given by Equation (A1) below.

$$\mathbf{R} = \begin{array}{c} \begin{matrix} u_{b,\,\text{node 1}} & u_{b,\,\text{node 2}} & \cdot & \cdot & \cdot & u_{b,\,\text{node } n} \end{matrix} \\ \begin{bmatrix} x_{(11)} & x_{(21)} & \cdot & \cdot & x_{(n1)} \\ \cdot & \cdot & \cdot & \cdot & \cdot \\ x_{(1j)} & x_{(2j)} & \cdot & \cdot & x_{(nj)} \\ \cdot & \cdot & \cdot & \cdot & \cdot \\ x_{(1k)} & x_{(2k)} & \cdot & \cdot & x_{(nk)} \\ \cdot & \cdot & \cdot & \cdot & \cdot \\ x_{(1n)} & x_{(2n)} & \cdot & \cdot & x_{(nn)} \end{bmatrix} \begin{matrix} \mathbf{x}_{i1} \\ \cdot \\ \mathbf{x}_{ij} \\ \cdot \\ \mathbf{x}_{ik} \\ \cdot \\ \mathbf{x}_{in} \end{matrix} \end{array} \qquad (\text{A1})$$

From the reference matrix **R**, only the rows corresponding to the respective measured coordinates are extracted. For example, considering $j$ and $k$ as the two measured coordinates, only the rows corresponding to those two coordinates (marked in dotted and thin-lined boxes respectively) are extracted from **R**.

$$\mathbf{R}_{\,extract} = \begin{array}{c} \begin{matrix} u_{b,\,\text{node 1}} & u_{b,\,\text{node 2}} & \cdot & \cdot & \cdot & u_{b,\,\text{node } n} \end{matrix} \\ \begin{bmatrix} x_{(1j)} & x_{(2j)} & \cdot & \cdot & x_{(nj)} \\ x_{(1k)} & x_{(2k)} & \cdot & \cdot & x_{(nk)} \end{bmatrix} \begin{matrix} \mathbf{x}_{ij} \\ \mathbf{x}_{ik} \end{matrix} \end{array} \qquad (\text{A2})$$

The rows $\mathbf{x}_{ij}$ and $\mathbf{x}_{ik}$ obtained from the FE model in Equation (A2) are divided by the corresponding measured maximum amplitude, $x_{m,j}$ and $x_{m,k}$ to obtain the relatively displacement as stated in Equation (11). This step and further explanations are covered in Section 2.2.1 and Figure 1.

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
