# Peer review of "Unbalance Estimation for a Large Flexible Rotor Using Force and Displacement Minimization"

_machines, doi:10.3390/machines8030039_

Round 1
Reviewer 1 Report
The paper compares a proposed balance method based on displacement methods with a force-based method from the literature. The methods are applied to guiding rolls for paper machine, which is indeed an interesting case. These rolls can have very complex unbalance distribution due to its nature (large tubes) and correction masses are placed at the ends of the rolls. This leads to a question about how realistic is the assumption that the rolls have a single point mass at a single position?
Overall the paper is interesting and well written. I have some suggestions for discussions that would enrich the paper. The only point that I really miss in the formulation of the methods is how the identified unbalance is translated to balancing masses to be added in the balancing planes after the unbalance identification.
- Is it possible to adapt the model described in section 2.2 for rotors with multiple discs (multiple unbalances to be identified)? From what is described in section 2.2.1, it seems that only one unbalance location is identified.
- Please add a description of the prediction algorithm used. Eq. 14 gives the objective function, but how was the optimization done? Some MATLAB routine like fminsearch?
- Line 229 (page 8): Missing a "to": "according to the method".
- In Figure 6 (page 12) it is not very clear why there is a time axis in the graphs. Do the graphs represent the rotor response of one cycle of the steady-state response? If yes, please mention it in the text.
- In section 4.2, only the vertical displacements were used for the identification. Did you try to do the identification with the horizontal displacements? Is there any reasoning behind choosing the vertical displacements or is it simply because they were better than using the horizontal displacements? Did you try to use both vertical and horizontal displacements? That would be an interesting discussion to add to the paper. The same comments are valid for section 5.2.2.
- From figure 7d, it seems that the method is very likely to lead to a completely wrong result when comparing the response of nodes 1 (0.5 variation) and 20 (0 variation). Did you do any investigation or have any comments regarding the robustness of the method?
- It would be nice to have a short discussion on why TSA was used instead of the more common frequency average method consisting of making several measurements and averaging their FFT spectrum.
- Line 301 (page 14): fix the plural form of the word "signal": "two simulated vibration signals".
- Line 417 (page 16): It seems that the "was" is not correct: "For cases 2 and 3, the method has identified".
- Line 420 (page 16): This line needs to be rephrased.
- Line 421 (page 17): I suggest using "the lowest".
- Line 432 (page 17): I suggest replacing "match the best" by "have a better match".
Reviewer 2 Report
Review Unbalance estimation for a large flexible rotor using force and displacement minimization
This is an interesting topic well worth studying. The proposed method seems to consist of an initial stage to identify a location where a single out of balance mass might be located to replicate the measured out of balance responses. By doing this, the resulting balancing can be performed more effectively. This is not necessarily a useful feature for many machines, where there may be a severe limitation to where balancing could be applied, but in the case of a long, open rotor such as the paper roller discussed, it is more likely that this could be exploited. There are some examples given numerically and also experimentally.
A few comments:
- Figure 1 doesn’t quite work as a flow chart - Flow chart doesn’t quite make sense – there seem to be two simultaneous entry points to step 2. Please clarify.
- When constructing reference cases (for eq 9), are multiple drive speeds considered? I can see this working if just a single target drive speed is considered, perhaps less well if multiple speeds that activate different mode shapes are considered.
- There are widespread references to taking the ‘maximum displacements’ - if this is a linear structure and responding harmonically, isn’t the maximum displacement simply the amplitude of the response? Wouldn’t this be a better term to use? Assuming it is an amplitude wouldn’t it be more efficient to use an analytical calculation of response based on the system matrices rather than the time series numerical simulation shown in figure 6?
- Figure 3 shows a lot of drive speeds interacting with the 1st two modes at subharmonics. Apart from the suggested cause of ‘n per rev’ excitation coming from bearings etc did you consider the potential for nonlinearity to be causing these?
I think if the above issues can be considered, this is acceptable for publication in Machines.
Reviewer 3 Report
I strongly recommend this paper for publication due to the clear comparison of various fault detection methods which are compared with each other both in simulations and experiments. All results are carefully described and well explained. I only recommend a very minor revision which accounts for the following facts:
1) At some positions definite and/or indefinite articles seem to be missing. Moreover, the English noun "data" is a plural word. So, the correct phrasing would be "the measured data ARE evaluated..." instead to "the data IS evulated..."
2) In the introduction as well as in the list of references, you mention neural network techniques. Is it possible to add a comment which states whether or how your proposed approach could be compared with such techniques?
3) Fault detection and diagnosis requires the property that a fault is not shadowed by a different influence factor. How can it be guaranteed in your setting, that for example two different faults occurring at the same time do not level out in such a way that the fault becomes undetectable? I assume that this could be a direction towards optimal design of experiment, for example, to design optimal speed profiles instead of using only constant velocity. Maybe, you could comment on such an issue as well in the outlook.
